



# OSSE for a sustainable marine observing network in the Marmara Sea

Ali Aydoğdu[1,2,3], Timothy J. Hoar[4], Tomislava Vukicevic[5], Jeffrey L. Anderson[4], Nadia Pinardi[2,6], Alicia Karspeck[4], Jonathan Hendricks[4], Nancy Collins[4], Francesca Macchia[7], Emin Özsoy[8]

[1] Science and Management of Climate Change, Ca' Foscari University of Venice, Italy

[2] Centro Euro-Mediterraneo sui Cambiamenti Climatici, Bologna, Italy

[3] Nansen Environmental and Remote Sensing Centre, Bergen, Norway

[4] National Center for Atmospheric Research, Boulder, Colorado, USA

[5] Office of Water Prediction, National Weather Service NOAA, USA

[6] Department of Physics and Astronomy, University of Bologna, Italy

[7] Barcelona Supercomputing Center (BSC), Barcelona, Spain

[8] Eurasia Institute of Earth Sciences, Istanbul Technical University, Turkey

Correspondence to: Ali Aydoğdu (ali.aydogdu@nersc.no)

**Abstract.** An observing system simulation experiment (OSSE) is presented in the Marmara Sea. A high resolution ocean circulation model (FESOM) and an ensemble data assimilation tool (DART) are coupled. The OSSE methodology is used to assess the possible impact of a ferrybox network in the eastern Marmara Sea. A reference experiment without assimilation is performed. Then, synthetic temperature
and salinity observations are assimilated along the track of the ferries in the second experiment. The results suggest that a ferrybox network in the Marmara Sea may improve the forecasts significantly. The salinity and temperature errors get smaller in the upper layer of the water column. The impact of the assimilation is negligible in the lower layer due to the strong stratification. The circulation in the Marmara Sea, particularly the Bosphorus outflow, helps to propagate the error reduction towards the western basin
where no assimilation is performed. Overall, the proposed ferrybox network can be a good start to design an optimal sustained marine observing network in the Marmara Sea for assimilation purposes.

## 1 Introduction

The Marmara Sea is one of the compartments of a water passage known as the Turkish Straits System
(hereafter TSS). The TSS connects the Black Sea and the Mediterranean by two narrow straits, namely the Bosphorus and the Dardanelles, along with the Marmara Sea.

Salty and dense Mediterranean waters and brackish Black Sea waters form a two-layer exchange flow through the TSS. In combination with the complex topography, flow structures in different scales generate





a challenging environment for oceanographic studies.

Until recently, the need for high resolution in the straits made it infeasible to model the complete TSS due to the computational cost. However, integral models of the system have emerged with increasing computational capacity in recent years [Gürses et al., 2016; Sannino et al., 2017; Stanev et al., 2017].

5     In this study, to the best of our knowledge, we present the first data assimilation experiments performed in the Marmara Sea. In the region, in-situ observations are generally scarce and collected by dedicated projects [Beşiktepe et al., 1994; Chiggiato et al., 2012; Özsoy et al., 2001; Tuğrul et al., 2002; Ünlüata et al., 1990] for a limited area and time. Moreover, the spatial resolution and frequency of satellite observations are still not sufficient to monitor the system continuously. For these reasons, a sustainable 10 marine monitoring network is required in the Marmara Sea. We propose a ferrybox network mounted on existing public transportation infrastructure.

We follow the OSSE methodology to achieve our goal. OSSE has been used widely in the atmospheric community for four decades for design of new observation tools, error assessment in large models and parameter estimation [Arnold Jr and Dey, 1986; Masutani et al., 2010]. Atlas [1997] summarizes the 15 criteria established by the atmospheric community to perform credible OSSE. Halliwell Jr et al. [2015, 2014] gave an example of an ocean OSSE in the Gulf of Mexico by following those criteria. Aydoğdu et al. [2016] studied a fishery observing system in the Adriatic Sea taking the same criteria into account.

This paper is organized as follows: In the next section, the main characteristics of the TSS relevant to this study are summarized. In section 3, the model and data assimilation scheme that we used are 20 documented. Section 4 is devoted to the OSSE design. The nature run and forward model are introduced. Ferrybox network design is also detailed and the methodology for impact assessment is outlined. The results are discussed in section 5. The summary and conclusions are presented in the last section.

## 2    Overview of the Turkish Straits System

The Marmara Sea with the Bosphorus and the Dardanelles Straits constitute the Turkish Straits System 25 (TSS) which connects the Black Sea and the Mediterranean. The exchange of contrasting water masses forms a highly stratified water column structure throughout the system. The brackish surface water of the Black Sea flows towards the Mediterranean in the upper layer and the salty and dense Mediterranean Sea water occupies the lower layer of the water column [Ünlüata et al., 1990]. In addition to the strong stratification, the complex topography of the two narrow straits and an internal sea with shallow shelves 30 and deep depressions presents a unique and challenging environment for oceanographic studies.

The Bosphorus is an elongated narrow strait connecting the Black Sea and the Marmara Sea. The upper layer flow is dominated by the Black Sea water with salinity about 18 psu. The lower layer water originates from the Mediterranean and reaches the Black Sea with a salinity of about 37 psu. The interface depth between the two layers is mainly determined by the salinity gradient. In summer, a three layer temperature





structure appears. A warm upper layer due to the atmospheric heat flux overlays a cold intermediate layer that propagates from the Black Sea. Warmer Mediterranean water constitutes the bottom layer [Altıok et al., 2012]. The constriction in the mid-section and the sill in the southern section of the strait apply a hydraulic control on the flow and produce a surface jet at the Marmara Sea exit [Sözer and Özsoy, 2017].

The velocity of the Bosphorus jet can exceed 2 m/s [Jarosz et al., 2011]. Therefore, the jet is one of the key factors influencing the dynamics of the Marmara Sea. It enhances the mixing by entraining lower layer water [Ünlüata et al., 1990], energizes the Marmara Sea circulation in the eastern basin [Beşiktepe et al., 1994] and produces mesoscale eddies due to the potential vorticity balance [Sannino et al., 2017].

The wind is another important factor influencing the dynamics of the Marmara Sea. Many strong

cyclones pass over the Marmara Sea especially in winter [Book et al., 2014]. Northeasterly winds dominate the atmospheric circulation throughout the year.

The third important dynamical constituent is the density-driven baroclinic flow in the lower layer [Hüsrevoğlu, 1998]. The velocity of the flow in the lower layer can reach 1 m/s [Jarosz et al., 2012].

The high complexity of the system requires integral modelling approaches to represent the links between

15 its different compartments.

# 3    Ensemble modelling and data assimilation in the TSS

The ocean model used in this study is the Finite Element Sea-ice Ocean Model (FESOM). FESOM is an unstructured mesh ocean model using finite element methods to solve the hydrostatic primitive equations with the Boussinesq approximation [Danilov et al., 2004; Wang et al., 2008]. It was developed by the

20 Alfred Wegener Institute as the first global ocean model using an unstructured mesh. Gürses et al. [2016] applied FESOM to the TSS and performed realistic simulations of the complete system using atmospheric forcing. The TSS model domain extends zonally from 22.5°E to 33°E and meridionally from 38.7°N to 43°N. The mesh resolution is as fine as 65 m in the Bosphorus and 150 m in the Dardanelles. In the Marmara Sea, the resolution is always finer than 1.6 km and is not coarser than 5 km in the Black Sea

and the Aegean Sea. The water column is discretized by 110 vertical z-levels. Vertical resolution is 1 m in the first 50 m depth and increases to 65 m at the bottom boundary layer in the deepest part of the model domain. The mesh has 70240 nodes at the surface layer and more than 3 million nodes for the 3D state variables i.e. temperature, salinity, zonal and meridional velocity.

Aydoğdu et al. [2018] performed inter-annual simulations using a similar model setup. Two six years

simulations using different surface salinity boundary conditions have been completed and evaluated. It is shown that the model is able to simulate main characteristics of the system qualitatively. It simulates the two layer structure of the TSS, successfully. Moreover, dynamical properties such as sea level difference between different compartments and dynamics related to high frequency atmospheric events are well-captured. However, the error growth in temperature and salinity throughout the simulation is also



demonstrated which motivates the need for data assimilation in the TSS.

In this study FESOM has been coupled with an ensemble-based data assimilation framework, Data Assimilation Research Testbed (DART). DART is an open-source community facility developed at NCAR [Anderson et al., 2009] that provides data assimilation tools to work with either low-order or high-order models for different research activities [Hoteit et al., 2013; Karspeck et al., 2013; Raeder et al., 2012; Schwartz et al., 2015].

DART includes several different stochastic and deterministic ensemble Kalman filtering algorithms. In this study, we use the Ensemble Adjustment Kalman Filter (EAKF) as described in Anderson [2001]. The EAKF is a deterministic ensemble Kalman filter, where the observations are not perturbed randomly before they are assimilated. One of the main advantages of the EAKF for our application is that it preserves the prior covariance information. In the TSS, where different dynamics compete and generate circulation structures at various scales, the prior information is crucial for maintaining the dynamical balances among different scales. In addition, the covariances are updated in every assimilation cycle which is important to sustain the high frequency variability of the system.

One of the main issues to take into consideration in ensemble data assimilation is the filter divergence. It can result from insufficient ensemble variance which leads to assigning more weight to the prior information which in turn may cause rejection of information from observations. As a result, the analysis departs from the observations [Anderson, 2001]. There are techniques to prevent filter divergence such as inflating the covariances [Anderson and Anderson, 1999]. In the standard DART, inflation of the prior ensemble leads to multiplying the prior covariances by a constant slightly larger than 1. Instead of DART's multiplicative covariance inflation, we have applied random perturbations to the background vertical diffusivity for each ensemble member since one of the main sources of error in the model is the vertical mixing and the position of the interface between the upper and lower layers. The background vertical diffusivity $\kappa_{v0}$ which is implicity represented in the $\kappa_v$ of the tracer equations is randomly perturbed all over the domain every timestep by fitting a gaussian with a mean of 0 and a standard deviation of 5% of the $\kappa_{v0}$.

Houtekamer and Mitchell [1998] shows that spurious correlations in the prior information can be avoided by using large ensembles. Since it is computationally expensive to use large ensembles in large geophysical models, they proposed a spatial localization technique to overcome spurious small correlations associated with remote observations. The impact of an observation on state variables is reduced as a function of distance from the observation with impact going to zero at a finite cutoff value. In DART, the cutoff value can be set as a constant value in radians. The localization function is the fifth order piecewise continuous, compactly supported as presented in Gaspari and Cohn [1999]. The horizontal cutoff radius, which is the half-width of the Gaspari-Cohn kernel, is used to deal with the high spatial variability of the water masses along the TSS. Moreover, the water column in the TSS is strongly stratified in the vertical. Therefore, vertical localization is also applied to update of the state in the upper layer of the water column




**Fig. 1.** Flow of FESOM/DART system. Reproduced after Anderson et al. [2009] with modifications for the TSS application

with less impact in the lower layer. As a result, the localization is an ellipsoidal volume centered on the observation being assimilated with a horizontal radius of two cutoff value and vertical radius of two cutoff value normalized by a factor. This approach is chosen not to face with problems, such as the breaking of





the stratification during the integration.

For interfacing FESOM and DART, a forward operator that maps model state vector into observation space, characterized by a type of physical quantity, geo-referenced location and time was defined. For the ferrybox type of observations we used a simple, but theoretically justifiable nearest-neighbor mapping. Specifically, for each discrete observation value the forward operator first finds the closest horizontal grid point in the state vector and then the closest vertical location.

The details of the FESOM/DART coupling are shown schematically in Fig. 1. The interface to DART requires two model specific routines that convert the model restart files to DART state vectors and vice versa. For the model to DART step the state vector is provided from each ensemble member. The updated state vectors are then converted to the model restart files as an initial ensemble for the next assimilation cycle within which FESOM is applied to integrate the model state forward to the next analysis time. This process is repeated until the experiment finishes.

# 4    Design of the OSSE

The OSSE methodology used is shown schematically in Fig. 2. A proposed ferrybox network was assumed to be in the eastern Marmara Sea using the existing transportation infrastructure. Therefore, we were able to the simulate the observations from the ferrybox network by tracking the ferrylines in the Marmara Sea. The impact assessment of this network was performed using the FESOM/DART ensemble data assimilation system.

## 4.1    The nature run and forward model

The OSSE methodology requires a representation of a reference true atmosphere or ocean which is called the nature run (NR). The NR is used for generating the synthetic observations to assimilate and for assessing the quality of the assimilation. For a fully objective OSSE the NR should be created using a simulation from a different model than the forward model (FM) used in the assimilation in an attempt to simulate the inevitable model errors.

In the current study the fraternal twin method was used, where the NR and FM are based on the two different configurations of the same model. The difference between the NR and the FM configurations is the surface salinity boundary condition, where the former employed the relaxation boundary condition for the sea surface salinity whereas the latter uses the mixed salinity boundary condition [Huang, 1993].

This approach was chosen because FESOM in the Marmara Sea is sensitive to equally plausible salinity boundary conditions. This was demonstrated through long-term simulations using both configurations, which were validated with the actual in-situ observations [Aydoğdu et al., 2018]. The simulations were performed for period 1 January 2008 to 31 December 2013. It is demonstrated that they deviate from





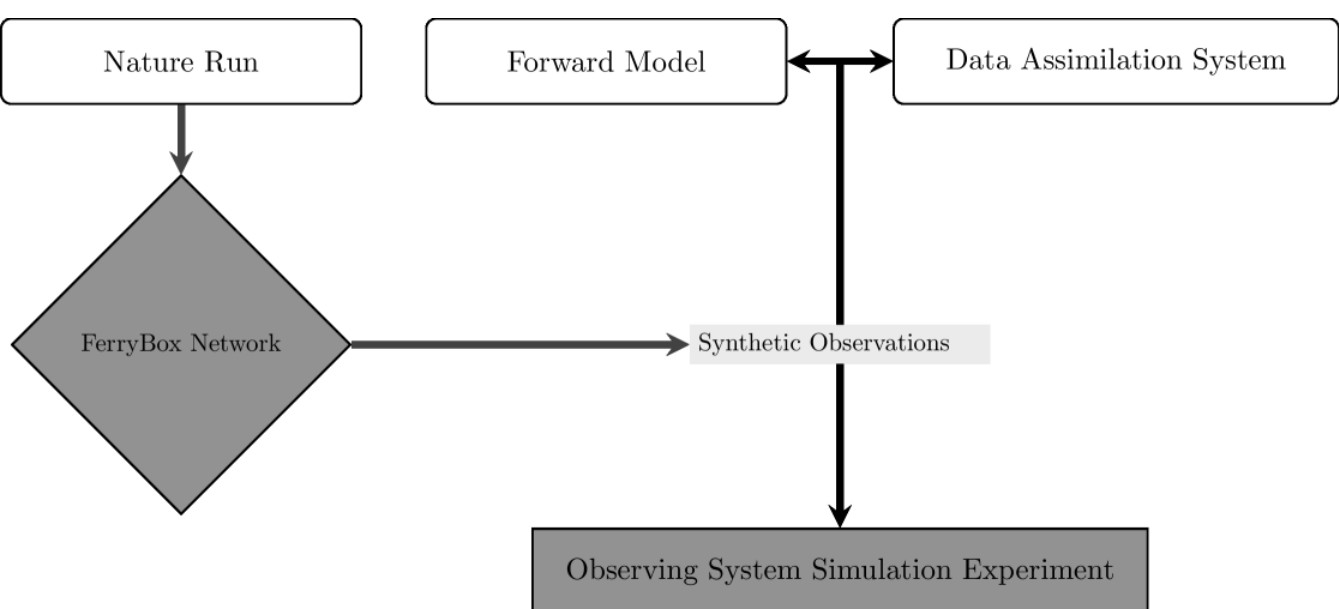

**Fig. 2.** OSSE methodology applied in the Turkish Straits System. The forward model (FM) is a different configuration of the model setup used for the Nature Run (NR) as detailed in the text.

each other but both are still realistic during the OSSE period which is 1-8 January 2009. The output of the NR was saved hourly during the OSSE period to generate synthetic observations.

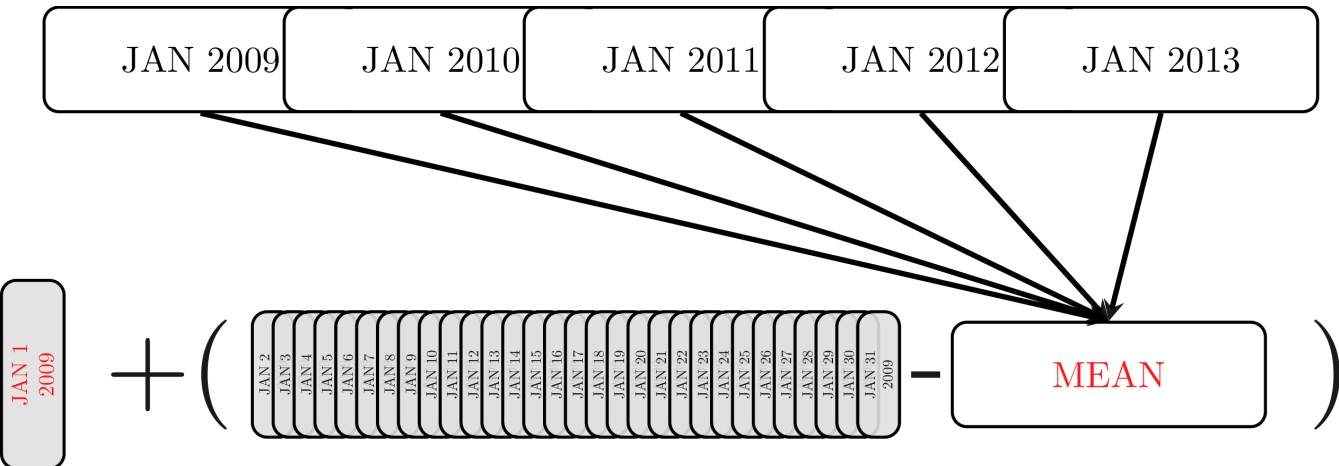

**Fig. 3.** Schematic representation of the methodology used to generate the initial ensemble. MEAN is the average of January over five years between 2009-2013

The ensemble consists of 30 members. An initial ensemble is produced for 1 January 2009 as schematized in Fig. 3. The five-year means of the temperature and salinity for January are computed from the inter-annual simulation. Then, the deviations of each day between 2-31 January 2009 from the inter-annual



mean are calculated. Finally, these deviations are added to the temperature and salinity fields of 1 January 2009 to provide an initial perturbation for each ensemble member. Fig. 4 shows the variance of the initial ensemble for salinity and temperature for the depths 5 m and 20 m. The horizontal distributions of the variance are similar for both variables. The variance is larger in the exits of the straits. In particular, the spread at the Bosphorus exit for the upper layer down to 15 m is larger due to the variability in the outflow. Correspondingly, the Dardanelles inflow to the Marmara Sea increases the variance in the lower layer. Such a distribution of initial variance in the Marmara Sea is appropriate to initialize the experiments since the assimilation of synthetic observations is performed in the impact area of the Bosphorus outflow where the ensemble spread can be diminished by the assimilation.

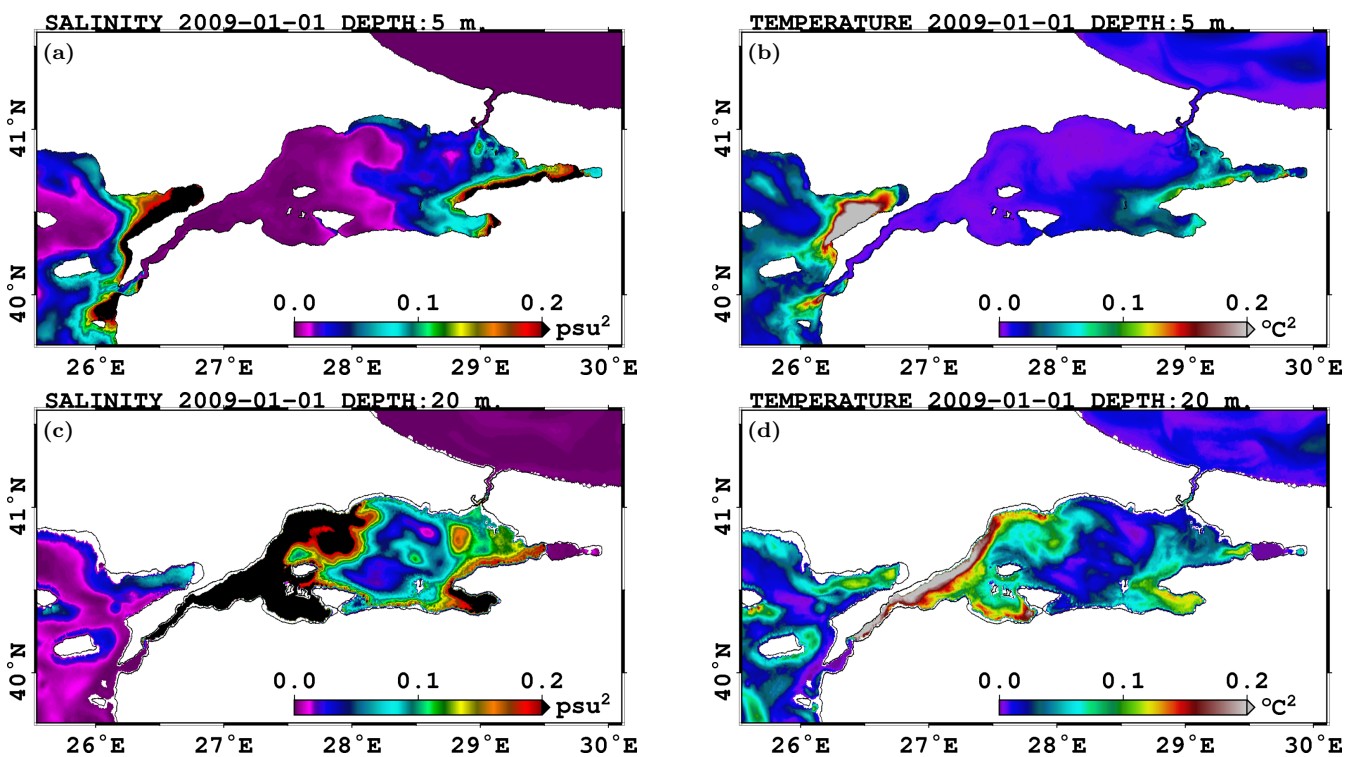

**Fig. 4.** Salinity (left) and temperature (right) variance of the initial ensemble at 5 m (top) and 20 m (bottom) depth on 1 January 2009 at 00:00 UTC.

## 4.2   The eastern Marmara Sea ferrybox network design

The Marmara region has the highest urban population in Turkey. It includes the metropolitan city of Istanbul which is divided into two by the Bosphorus and surrounded by the Black Sea in the north and the Marmara Sea in the south. As a consequence, a network of ferries is an essential means of transportation.





The main hub for the ferries is Istanbul which is connected to the other cities around the Marmara Sea by several ferries (Fig. 5a).

In the Marmara Sea, it is not easy to deploy instruments such as argo or glider since there is heavy ship traffic. However, the infrastructure for a ferrybox network is already available. The ferrylines in the Marmara Sea cover most of the eastern basin including the Marmara Sea exit of the Bosphorus. Using the ferry network as a monitoring system would be an efficient way to build a sustainable ocean observing network in the Marmara Sea.

Some existing state-of-the-art ferrybox networks are discussed by Petersen [2014]. Usage has increased since the European network for ferrybox measurements project[1], especially in the northern European seas. In the Mediterranean, there is a ferrybox system between Piraeus and Heraklion operated by HCMR [Korres et al., 2014]. A ferrybox is mainly includes temperature, salinity, turbidity and chlorophyll-a fluorescence sensors, and a GPS receiver for measuring position. Oxygen, pH, $pCO_2$ or algal groups as well as air pressure, air temperature and wind sensors can also be installed [Petersen, 2014]. The ferrybox observations can be used for analyzing the state of the ocean [Seppälä et al., 2007], comparison with other instruments [Sørensen et al., 2007] and can also be assimilated to improve the state of the ocean models [Grayek et al., 2011].

The sampling rate of the data can can vary between systems. The data used in Grayek et al. [2011] is sampled at 10 s intervals. For our OSSE, we use a sampling rate of 1 min. following Korres et al. [2014]. The synthetic observations are obtained from hourly NR outputs at varying spatial location along the track of the ferries. The depth of the sampling is set to 5 m, an average depth given by Grayek et al. [2011]. A random error, sampled by a gaussian around zero mean and standard deviation of $0.1°C$ for temperature and 0.04 psu for salinity, is added to each synthetic observation following Aydoğdu et al. [2016] in order to simulate realistic measurements taking the instrumental error into account. These synthetic observations are the same for each ensemble member i.e. the kalman filtering is not stochastic. On the other hand, observational error matrices in the Kalman gain are chosen to be constant diagonal with $0.5°C$ and 0.25 psu for temperature and salinity as proposed by Grayek et al. [2011] considering other sources of error such as representativeness.

Three different ferry routes are chosen from the map in Fig. 5a, and their tracks are approximated by observing from the real-time Marine Traffic[2] application (Fig. 5b). The longest duration for a cruise in the eastern Marmara Sea is about 3.5 hours between Ambarlı-Topçular (Table 1). Another transect used here is YeniKapı-Yalova which takes about 75 min. and has cruises every two hours from each port. This route directly crosses the Bosphorus outflow and has the highest number of cruises a day. Therefore, we include six ferries in various periods of the day for YeniKapı-Yalova transect. The last transect chosen is Yenikapı-Bandırma crossing the Marmara Sea from north to south. The navigation on this line takes

---

[1] http://www.ferrybox.org
[2] https://www.marinetraffic.com





| Route | Location | Distance | Speed | Duration | Departure Time |
|---|---|---|---|---|---|
| Yenikapi | 28.956E-41.002N | | | | 09:45,15:45,21:45 |
| \| | | 46.2km (28.7mi.) | 23 kn | 75 min. | |
| Yalova | 29.274E-40.661N | | | | 07:15,13:45,19:45 |
| | | | | | |
| Yenikapi | 28.956E-41.002N | | | | 07:30 |
| \| | | 110.0km (68.5mi.) | 27 kn | 150 min. | |
| Bandirma | 27.967E-40.354N | | | | 18:30 |
| | | | | | |
| Ambarli | 28.676E-40.966N | | | | 20:00 |
| \| | | 70.0km (43.2mi.) | 12 kn | 210 min. | |
| Topcular | 29.434E-40.690N | | | | 16:00 |

Table 1: The unidirectional ferry tracks used in this study. The locations, distance between the ports, speed of the ferries, the duration of the cruise and time of departure from each port are listed.

about 2.5 hours. This transect is the only one that has direct impact in the southern basin. The resulting synthetic temperature and salinity observations sampled from the NR for the first day are shown in Fig. 6.

## 4.3 Experiments

Two experiments are performed as an initial evaluation for the data assimilation studies in the Marmara
Sea (Table 2). The first experiment FB001 is a reference experiment without assimilation. It is used to evaluate the errors when the synthetic observations are not assimilated. In the second experiment, FB002, all the synthetic observations are assimilated in the corresponding assimilation window. A six hours width is chosen for each assimilation window, since the area is under the influence of the Bosphorus jet which may develop high frequency variability in the water mass structure and circulation at the upper layer,
especially during severe storms and atmospheric cyclone passages.

| | Start Date | End Date | Assimilation | A.Cycle | Evaluation |
|---|---|---|---|---|---|
| FB001 | 01-JAN-2009 00:00 | 8-JAN-2009 00:00 | NONE | N/A | YES |
| FB002 | 01-JAN-2009 00:00 | 8-JAN-2009 00:00 | ALL | 6 hr | YES |

Table 2: Summary of the OSSEs. Start and end date of both experiments are the same. There is no assimilation in FB001 whereas all the data are assimilated in FB002. Assimilation cycle is 6 hr for FB002.

The horizontal cutoff radius is set to 6.36 km and the vertical cutoff radius is considered as 15 m centered on the observation location. The temperature and salinity increments after the first and third assimilation cycle are shown in Fig. 7a and Fig. 7b, respectively. As can be seen from the temperature corrections after the first cycle, the whole track of the Yenikapı-Bandırma route is not assimilated at once
since the southern section of the data is not in the current assimilation window. Moreover, the updates on the salinity fields are smaller around 20 m (Fig. 7b), compared to the upper layers (not shown) closer to the observation locations.



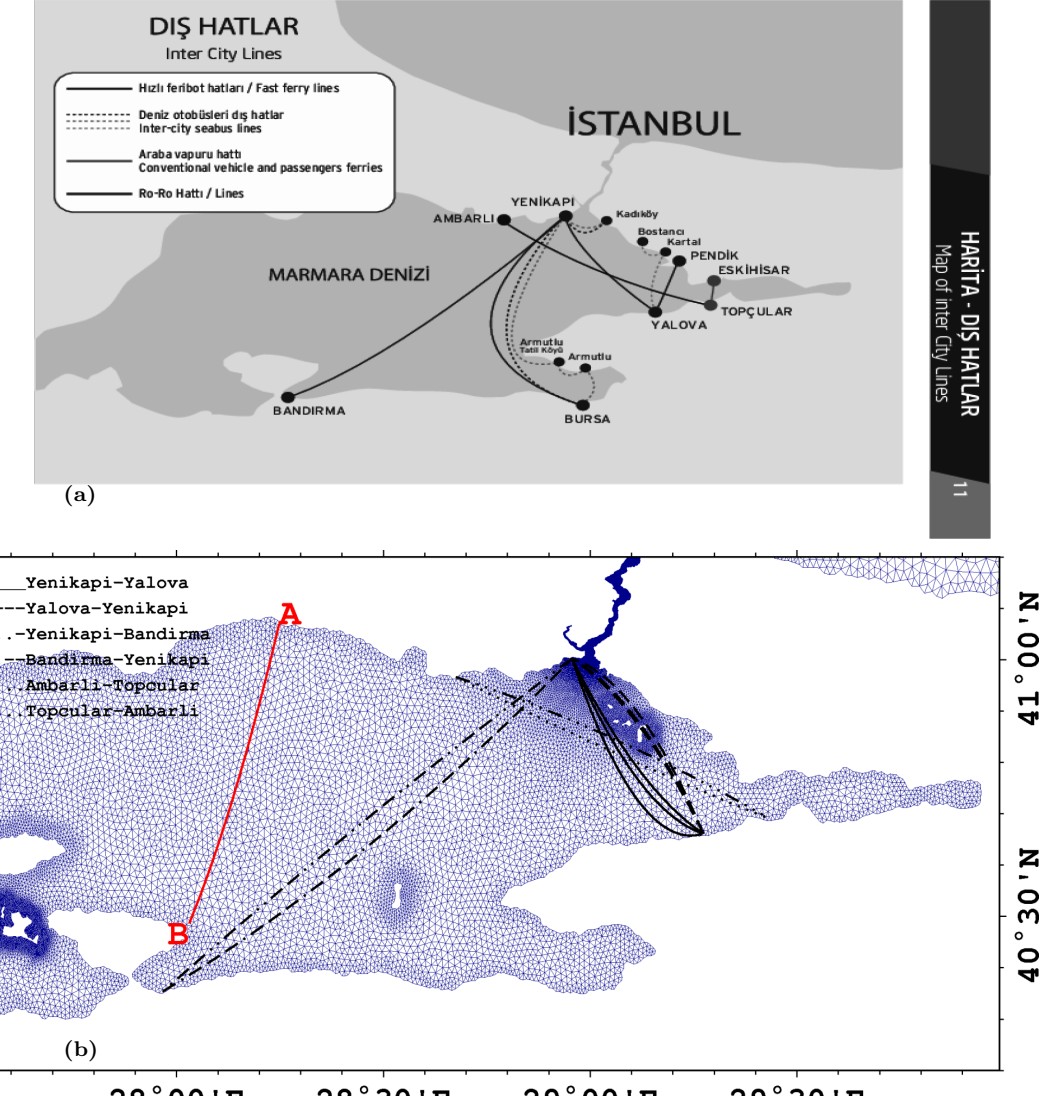

**Fig. 5.** a) The routes of the intercity ferrylines from Istanbul and to Istanbul suggested by the operating company IDO (http://www.ido.com.tr). b) Approximate unidirectional ferry tracks. The legend shows the direction of the ferries. The section A-B is used only for impact assessment against the NR. Triangular mesh of the model is underlaid.

## 4.4 Methodology for impact assessment

The DART offers tools to have the control on the data without any difficulty. It allows to decide to use a set of data for assimilation or evaluation. It also checks if any operator fails during the mapping of





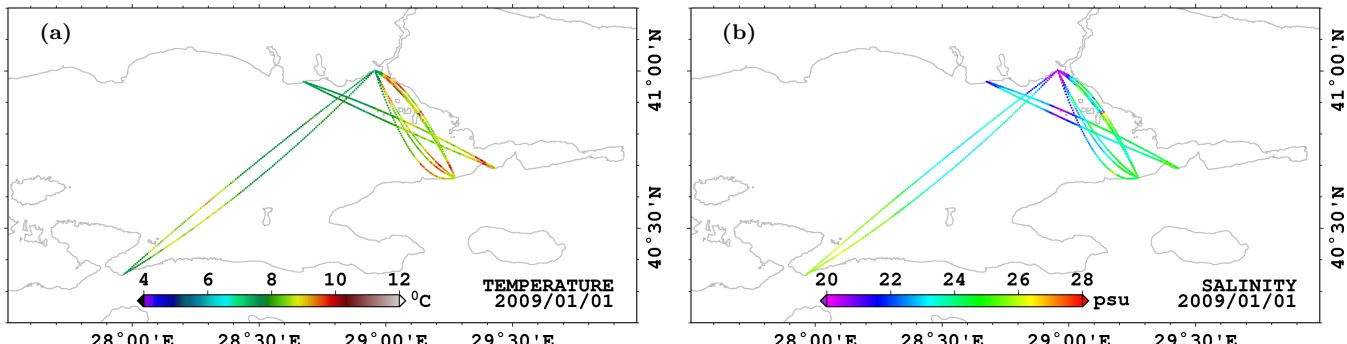

**Fig. 6.** Synthetic a) temperature and b) salinity observations on 1 January 2009 sampled from the NR.

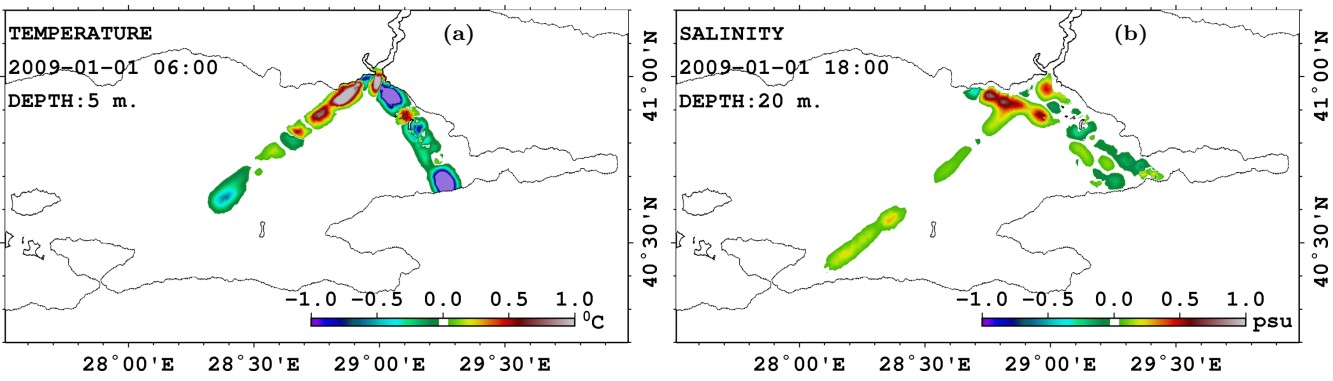

**Fig. 7.** Increments after assimilation for a) temperature at 01-01-2009 06:00 at 5 m depth and b) salinity at 01-01-2009 18:00 at 20 m depth.

the state vector to the observations. Synthetic observations are not used in the assimilation (rejected) if $x_b - y > TE(x_b - y)$. Here, $y$ is the observation and $x_b$ is the corresponding prior mean of the ensemble of forward operators. The expected value, $E$, is computed as:

$$E(x_b - y) = \sqrt{\sigma_{x_b}^2 + \sigma_y^2} \qquad (1)$$

where $\sigma$ stands for the standart deviation. T is chosen as 3 for both temperature and salinity in these experiments.

5    The first diagnostic we use to assess the impact of the observations is the RMS of innovations which are the root mean square of the difference between the prior and observation. We also use horizontal maps of the innovations to determine the spatial distribution of error.

The OSSE methodology allows various ways to evaluate the analysis since the NR is assumed to be the true state of the system. We exploit this assumption to compare the experiments with the NR to 10   understand the impact of assimilation better.

One diagnostic we use in this sense is to compare the NR and ensemble mean in the vertical. Although




we don't assimilate any data below 5 m and we limit the radius for vertical updates it is important to assess the vertical distribution of the errors given the strongly stratified water column in the Marmara Sea. For this purpose, we compute the difference between prior ensemble mean and NR along the transect A-B (see Fig. 5) in the first 50 m depth.

Second diagnostic is the RMS of difference between the prior ensemble mean and the NR computed in the first 10 m of the water column for the whole basin. The propagation of the error reduction can be traced from the spatial maps of this diagnostic.

# 5 Results

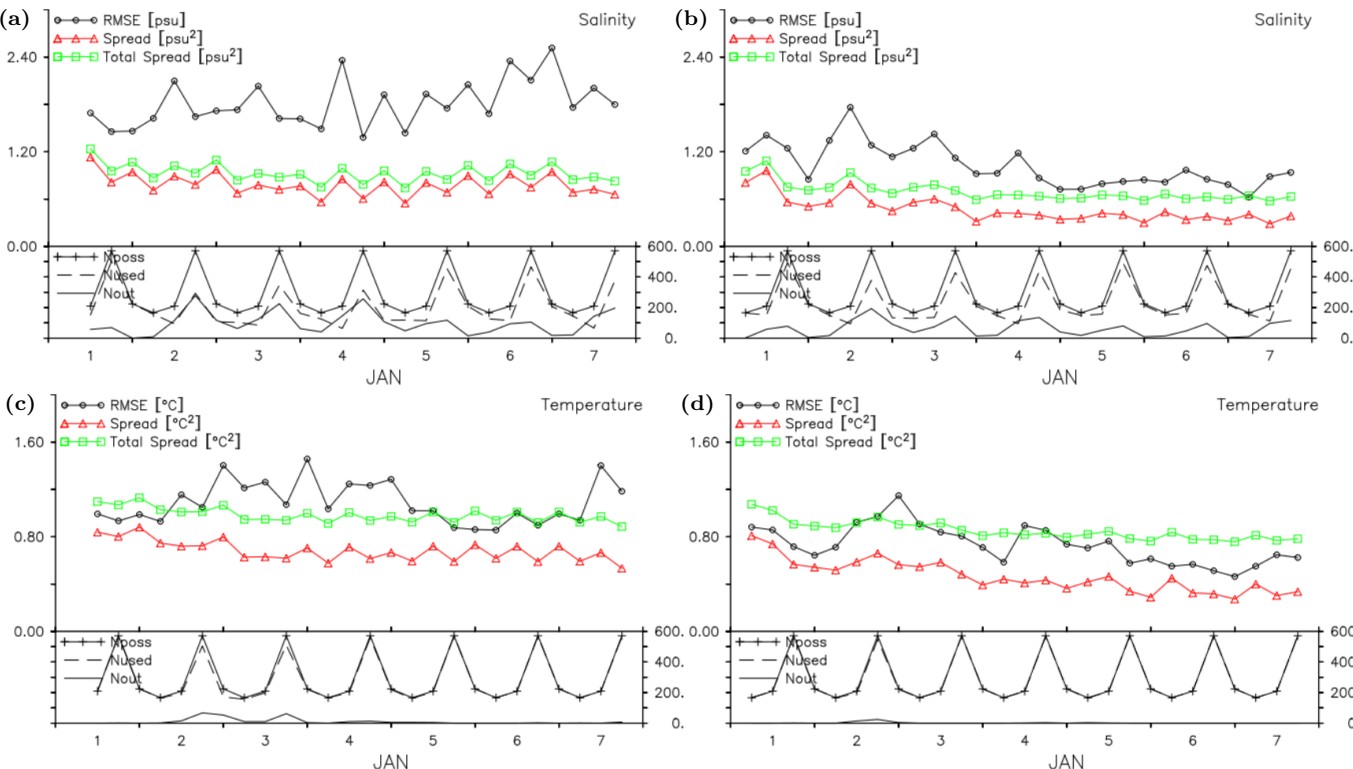

**Fig. 8.** Timeseries of RMS of innovations, spread and total spread of salinity (top) and temperature (bottom) for FB001 (left) and FB002 (right). The spread is the variance, therefore it is the square of the unit of the corresponding state variable. Total spread is the observational error added to the spread. Y-axis shows the range of each statistics indicated in the legend. Bottom panel of each figure shows the number of available observations (Nposs) in each assimilation cycle. For the assimilation experiment, FB002, Nused and Nout show the number of assimilated observations and outliers, respectively. For the experiment without assimilation, FB001, they are the number of observations which would be assimilated or rejected, respectively, in that specific assimilation cycle if assimilation was performed.



**Nonlinear Processes**
**in Geophysics**
Discussions



Figure 8 shows the time evolution of RMS of innovations, ensemble spread and total spread for temperature and salinity. The RMS of salinity innovations continuously grows in FB001. It fluctuates around 2 psu and reaches to 2.4 psu at the end of sixth day. In FB002, assimilation of the observations decrease the RMS of innovations, significantly. The RMS of innovations is generally below 1.2 psu. Although there is an increase of error in the first two days a gradual reduction takes place in the following days. The
RMS of temperature innovations is similar to that of salinity after the second day. The error grows in FB001 even though the trend is not as obvious as in salinity errors of the same experiment. The analysis is improved at the end of experiment FB002 compared to FB001. Another important result is that the ensemble still has spread comparable to the RMS errors at the end of the experiments (larger in FB001). In other words, the ensemble didn't collapse after a week of assimilation. We recall that the ensemble
spread is maintained by perturbing the background vertical diffusivity. The method seems promising at least for a week long period.

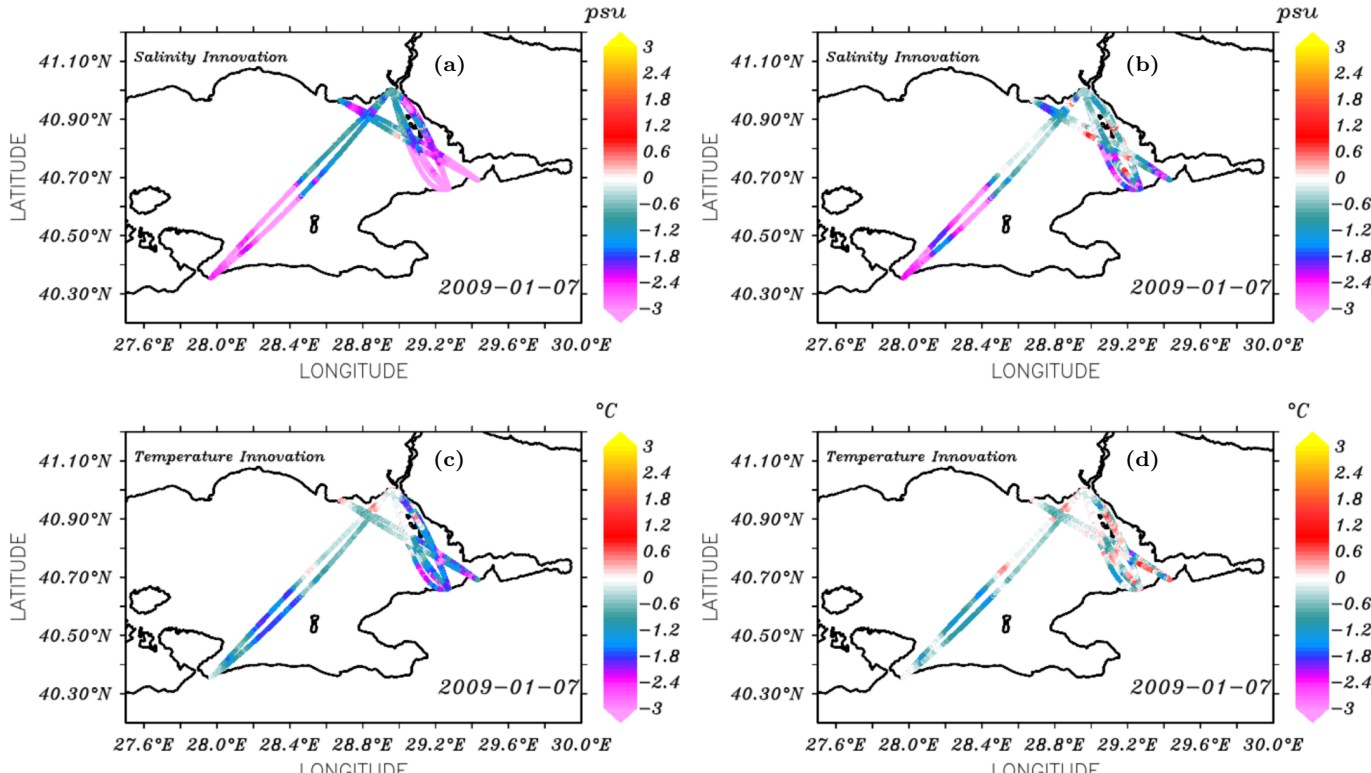

**Fig. 9.** Horizontal distribution of salinity (top) and temperature (bottom) innovations along the ferry tracks in 7 January 2009 for FB001 (left) and FB002 (right). Observations with a innovation out of the range ±3 are considered as outliers.

As discussed in section 4.4, the data are subjected to a quality control before assimilation. The bottom panels of Fig. 8 shows the number of available observations (Nposs), number of used observations (Nused)





and number of outlier observations (Nout). The decrease in the number of outlier observations in FB002 points out an improvement also in the regions in which the innovations are larger as can be deduced from Fig. 9.

In the last day of experiments (Fig. 9), the salinity innovations are better almost everywhere in FB002. There is a significant reduction in errors in the northern basin. Assimilation decreases the number of outlier salinity observations especially on the route between Yenikapı and Yalova. Moreover, innovations are also improved to a lesser extent in the southern basin where fewer observations are assimilated. The number of outlier temperature observations is very small in both experiments. Overall, the assimilation of temperature and salinity observations in the selected transects notably helps to correct the subsurface fields.

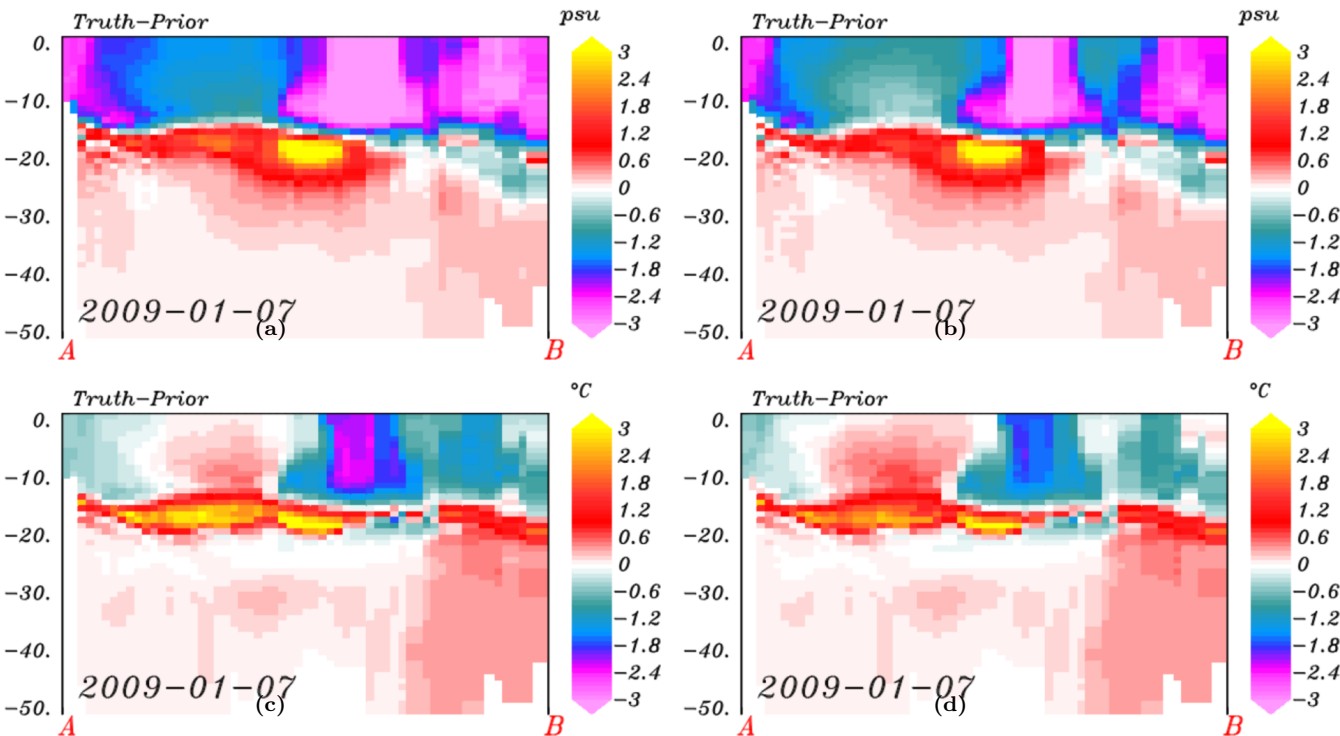

**Fig. 10.** Vertical distribution of salinity (top) and temperature (bottom) difference between the NR and the prior ensemble mean along the cross-section A-B (see Fig. 5) in 7 January 2009 for FB001 (left) and FB002 (right).

The improvement of the analysis is also noticed in remote areas such as the A-B transect in the central basin (see Fig. 5). Figure 10 shows the difference between the truth and prior state down to 50 m depth along the A-B transect after the last assimilation cycle. Comparison of salinity differences in FB001 and FB002 reveals the improvement in the northern section down to 15 m depth. The southern part away from the coast also gets better. The middle of the transect still has large discrepancies in both experiments at





the end of seven days. The correction in the remote area suggests a mechanism related to the outflow of
the Bosphorus and the surface circulation of the Marmara Sea. The water masses which are corrected by
assimilation in the eastern basin are pushed towards the west and reduce the error.

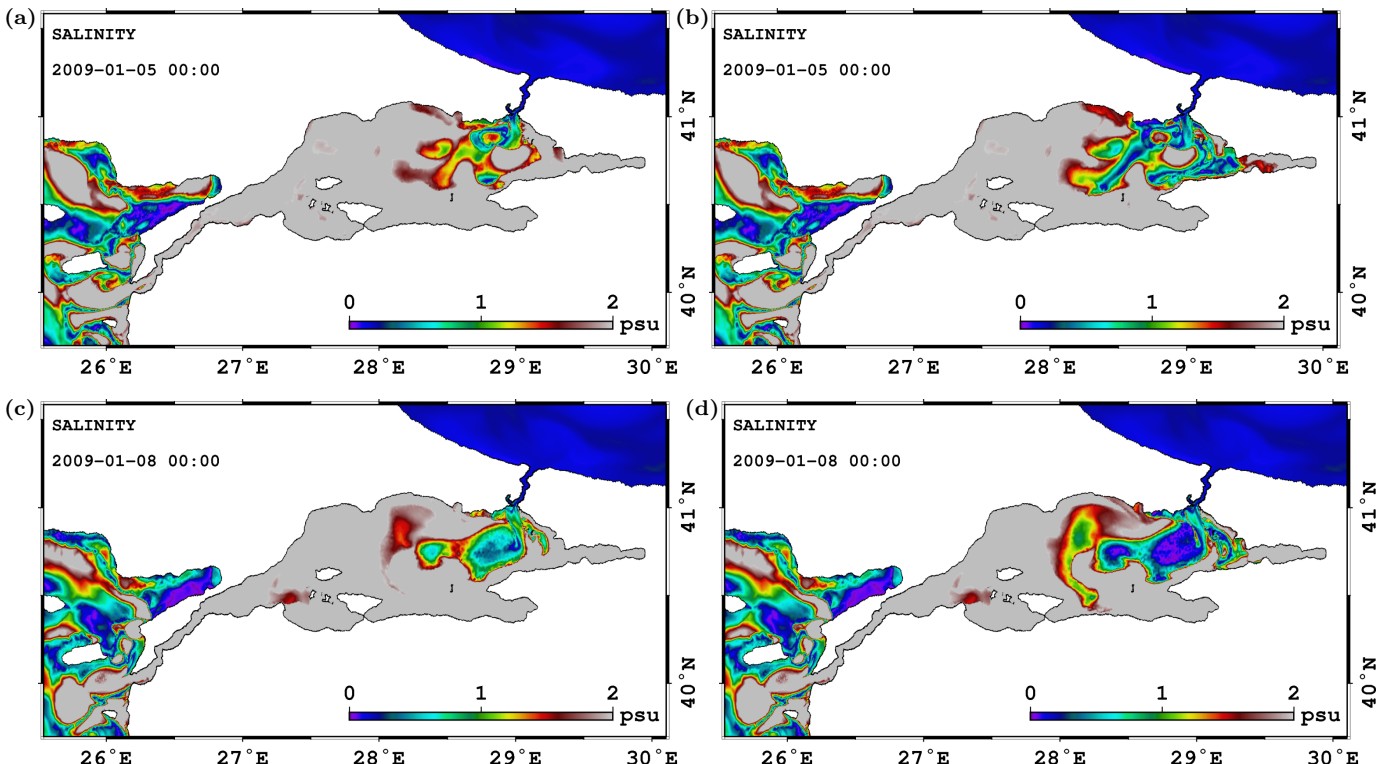

**Fig. 11.** RMS of the difference between NR and prior salinity at the first 10 m. Comparison of FB001
(left) and FB002 (right) are shown for 5 January 2009 (top) and 8 January 2009 (bottom). RMS of
difference is higher than 2 psu in the gray areas.

Figure 11 depicts the RMS of the difference between NR and prior salinity at the first 10 m depth.
It clearly supports the mechanism suggested above. The distribution of RMS of differences in the non-
5 assimilation case shows that the Bosphorus outflow also has some capability to reduce the difference since
the Black Sea water masses govern the upper layer of the Marmara Sea (Fig. 11a and 11c). These two
snapshots from the fifth and the last day of the FB001 show the error reduction in some regions due to the
dynamics. Therefore, the conclusion that the improvement is due to the assimilation is not straightforward.
However, comparison of the FB001 and FB002 reveals the role of the assimilation of ferry tracks on the
10 error reduction, clearly. In the fifth day, lower RMS of differences in FB002 (Fig. 11b) extend towards
the Gulf of Izmit in the east which is absent in FB001 (Fig. 11a). The westwards propagation is more
pronounced in the last day in FB002 (Fig. 11d) compared to FB001 (Fig. 11c). The central basin has
lower errors almost everywhere in the north-south orientation including the A-B transect shown before.



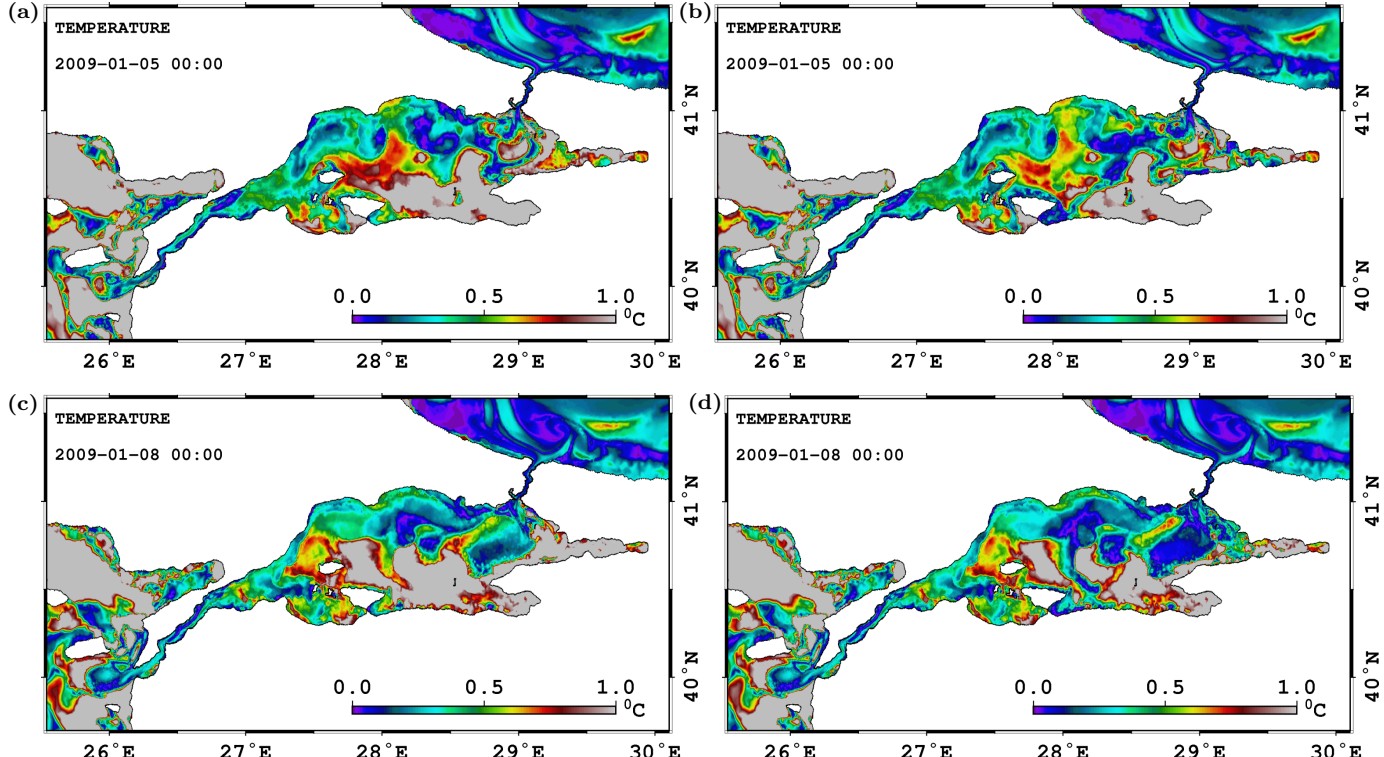

**Fig. 12.** RMS of the difference between NR and prior temperature at the first 10 m. Comparison of FB001 (left) and FB002 (right) are shown for 5 January 2009 (top) and 8 January 2009 (bottom). RMS of difference is higher than 1°C in the gray areas.

Finally, temperature fields are already closer to the truth in the fifth day even in the western basin (Fig.12). After seven days of assimilation, the temperature error in the Bosphorus plume is significantly reduced. The southern and central basin has improvements as much as 0.5°C locally.

# 6 Summary and Discussion

We have described data assimilation experiments performed in the Marmara Sea. The main characteris-
5    tics of the TSS have been summarized. For the study, a general ocean circulation model, FESOM and an ensemble data assimilation framework, DART have been coupled. The implementation of the data assimilation scheme has been reported.

The TSS is an important water passage for the oceanography of the neighboring Black and Aegean Seas. It also has important impacts on their ecosystem by maintaining the exchange of water masses and
10    nutrients. The high population in the cities surrounding the TSS and intense marine traffic through the passages add social and economic reasons to monitor the TSS in a sustainable way.





Real observations in the TSS have been obtained by dedicated projects for short time periods or have limited spatial coverage. Moreover, the satellite measurements are still low-resolution for monitoring and assimilation purposes. In this study, we proposed a sustainable marine monitoring network using the ferrylines in the eastern Marmara Sea. We think that equipping the ferries which operate daily everyday from Istanbul to various cities around the Marmara Sea with temperature and salinity sensors can provide immense amounts of data both in time and space. Given this motivation, we tested a ferrybox network including some of the ferry transects in the basin.

The OSSE methodology has been used to assess the impact of ferrybox measurements. We tried to satisfy the main criteria determined by approximately forty years experience of the atmosphere and ocean communities. However, it was still not possible to perform an OSE using real observations to compare with OSSE due to the lack of data during the experiment period.

The results of the two experiments presented here are promising. We showed that the assimilation of the salinity and temperature observations significantly improve the analysis in the Marmara Sea. The Bosphorus jet has an important role in the propagation of the error reduction towards the western basin where no data is assimilated. Moreover, the Marmara Sea circulation helps to improve the southern basin even for short timescales. The lower layer doesn't show any response to assimilation since a vertical localization around the observations around 5 m is applied to keep the impact in the upper layer as much as possible. Moreover, the stratification between the upper and lower layers is too strong so that it prevents the interaction between the two layers.

In conclusion, the results encourage further data assimilation studies in the Marmara Sea. The investigations can be extended to different observing systems, different areas of the sea or different dynamical focuses. Moreover, we believe the unique dynamics of the system demonstrated its ability to be a good natural laboratory for future data assimilation studies.

## Acknowledgement

This study is a part of the PhD thesis of Ali Aydoğdu. He is funded by Ca' Foscari University of Venice and CMCC during his PhD and REDDA project of the Norwegian Research Council during his Post-doc research period. The National Center for Atmospheric Research (NCAR) is Federally Funded Research and Development Center, sponsored by NSF.

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
