# Peer review of "OSSE for a sustainable marine observing network in the Marmara Sea"

_Nonlinear Processes in Geophysics, 2017_

## Referee Comment (RC1) · Anonymous Referee #1 · 16 Feb 2018

The manuscript presents an OSSE study in the Marmara Sea. The study is interesting for publishing in the Nonlinear Processes in Geophysics, because it provides an insight on how combining models and observations may improve the monitoring of the Marmara Sea characterized with complex dynamics. I have some important comments that should be addressed by the authors before accepting the manuscript for publication.

Major comments

1. Temperature and salinity observations are assimilated and the evaluation of impacts is made only with temperature and salinity. The assimilation of temperature and salinity observations, however, impacts all other model fields. The evaluation of the observational impacts should be extended at least to currents.

2. It seems that using the Ferrybox system for observing temperature and salinity near the surface is more feasible than using floats or gliders. On the other hand, does the improved estimate of temperature and salinity fields near the surface significantly improve the support for the most important applications of oceanographic forecasts in the Marmara Sea?

3. There are many technical details about specific solutions implemented in the data assimilation scheme, but impacts those solutions are not tested in the study.

4. I think that the style of writing should be improved. It is very difficult to read and interpret many sentences providing important information.

Minor comments

1. Page 2, line 2: The "high resolution" of what?

2. Page 2, line 3: What are "integral models"?

3. Page 2, line 12: The OSSE abbreviation is introduced, but it is not explained.

4. Page 3, line 14: same as comment 2.

5. Page 3, lines 19-20: This sentence is not related to the scope of the study.

6. Page 4, line 11: What is "covariance information" in this context?

7. Page 4, line 12: What is "prior information" in this context?

8. Page 4, line 13: Which covariances are updated?

9. Page 4, lines 20-24: Is vertical diffusion the most important for correctly simulating the depth of the interface? Vertical diffusion should be governed by slowly varying large-scale fields and perturbing it at the high frequency may add processes that may be unphysical.

10. Page 4, line 35: Localization may introduce strong dynamical imbalances. This contrasts the sentence on lines 11-13.

11. Page 6, lines 29-30: I do not understand this sentence.

12. Page 9, line 3: Argo floats and gliders are not instruments.

13. Page 9, lines 18-20: How is the sampling rate of 1 minute obtained output frequency of 1 hour.

14. Page 9, line 24: What is the meaning of stochastic in this context?

15. Page 9, line 25: Do you want to say that errors of observations are uncorrelated?

16. Page 10, line 16: Updates are smaller than what? It looks like temperature and salinity are compared by magnitude, but they are two different physical parameters.

17. Page 11: The bottom paragraph should be reformulated by using the correct terminology.

18. Page 12, lines 1-4: I do not understand why there are outliers in an OSSE experiment? Are observations wrong, or some assumptions are not valid?

19. Page 15, line 4: Innovations are better than what?

20. Fig. 11: Salinity differences have very large gradients. I suspect that they form strong density gradients impacting currents. Currents should be included in the evaluation of the assimilation.

---

## Referee Comment (RC2) · Anonymous Referee #2 · 8 Mar 2018

The system gave me access to the previously submitted other review. I agree with their judgment that this is an interesting paper and that it is eligible for publication in NPG eventually. I also agree with their major comments 1 and 2 (which are probably linked to each other). I do not agree with their major comments 3 and 4: (3) not testing the impact of all choices on the results does not appear to be a problem to me if those choices are clearly stated; (4) the style is generally good, with some exceptions (some of them given by the other reviewer in "Minor comments"). Please find below comments of my own.

Major comments:

A. Unless this has been published elsewhere, it would be useful to have a brief analysis of the physical situation in the NR at the time of the assimilation experiments, if possible

with figures.

B. I had an overarching question in my mind throughout my reading of the manuscript: is the combined effect of i.c. perturbations (a mix of short-term time lag and interannual variability) and diffusivity perturbations able to explain at least part of the model-data differences, within observational error? This question can be posed for (B1) simulated data, and (B2) any existing real observations (e.g., SST). The B1 question is a question of consistency of the innovations with Ensemble spread + obs error: it is partially covered in the ms. (e.g. Fig.8), but not exploited. The B2 question is about the realism of errors: it was not covered in the ms (only the conclusion mentions "lack of data"). Coming back to question B1: I have been frustrated that Fig.8 shows RMSE (of the dimension of the variable) and spread (of the dimension of the variable, *squared*). It would have been better to show *MSE* (not RMSE) and spread (assuming that this is *prior* Ensemble spread): then you could have tested whether the (prior) innovation variance was more or less of the same order as the (prior) "Total spread" (= your estimate of prior error + your estimate of obs error). I did the squaring visually, and the orders of both quantities do not match each other, especially for salinity. I believe that this should even briefly be discussed.

C. Why did you limit yourself to 7 days? Some of the error processes, especially those associated with mixing and stratification, could act on longer time scales.

D. The localisation cut-off scales are very short. Can't this trigger fast unphysical responses, for instance via temperature-sea level covariances?

Minor comments:

(some already made by the other Reviewer, but with my own words)

1. Page 4, lines 11-12: How is "prior covariance information" related to "dynamical balances"?

2. Page 4, lines 11 and 13: "the covariances are updated in every assimilation cycle":

Isn't this contradictory with "it preserves the prior covariance information"?

3. Page 4, line 25: The generally adopted procedure to perturb diffusivity parameters does not use a centered Gaussian pdf.

4. Page 14, line 5: "is similar" -> "behaves similarly"

5. Page 15, lines 8-9: "...correct the subsurface fields": Fig. 9 is in data space (surface), so one cannot see a subsurface effect from that figure.

---

## Author Comment (AC1) · 27 Apr 2018

**"OSSE for a sustainable marine observing network in the Marmara Sea"**

Aydoğdu et al., 2018, in revision, submitted to *Nonlinear Processes in Geophysics*

**Authors' responses**

We thank to the editor and the anonymous reviewers for comments motivating a revision of our paper which we are ready to perform based on their suggestions. We provide point-by-point responses to the reviewers' comments in the following, supplying the proposed updates and with few additional figures. Throughout this document, bold and italic fonts are used for the captions and reviewers' comments, respectively. Our responses are in normal fonts.

**Authors' response to RC#1**

**Major comments**

1. *Temperature and salinity observations are assimilated and the evaluation of impacts is made only with temperature and salinity. The assimilation of temperature and salinity observations, however, impacts all other model fields. The evaluation of the observational impacts should be extended at least to currents.*

In order to discuss this point, we provide a figure (Fig. 13) comparing salinity overlaid with current fields at 5 m. depth, obtained from experiments FB001 (left) and FB002 (right) for the exemplary case on 7 Jan 2009 at 00:00. The salinity differs significantly between the two experiments especially along the southeastern coast, while there is very little change in both qualitative and quantitative terms in the horizontal circulation, namely the current speed and direction, in the affected region that can be attributed to data assimilation. The effect of data assimilation is more pronounced in terms of the property fields, which alternatively indicates changes in stratification and vertical mixing along the southern coast.

Moreover, the same figure, but for the nature run is provided in the appendix as requested by the major comment A of the Reviewer#2. The circulation in the nature run (Fig. A2, introduced later) appear more intense compared to FB001 and FB002, however, without resulting in a significant change in the horizontal circulation patterns.

We can deduce that the impact of the assimilation on the circulation seems very small, if not negligible, compared to the differences of the experiments with the nature run.

[Figure]

**Fig. 13**. Comparison of the 5 m. salinity fields on 7 January 2009 at 00:00 between FB001 (left) and FB002 (right). The corresponding circulation patterns are shown by current vectors.

2. *It seems that using the Ferrybox system for observing temperature and salinity near the surface is more feasible than using floats or gliders. On the other hand, does the improved estimate of temperature and salinity fields near the surface significantly improve the support for the most important applications of oceanographic forecasts in the Marmara Sea?*

We agree with this comment in general. We will introduce a small paragraph in the revised text, explaining the needs and our rationale as expressed in the following.

The Marmara Sea has unique dynamics of its circulation, generated by volume fluxes through the straits, interaction with the atmosphere and buoyancy effects in a strongly stratified environment. All these factors play crucial roles in the dynamical response of the system. Black Sea and Aegean Sea water masses transported through the Marmara Sea determine its vertical structure, which in turn impacts its internal dynamics. In principle, all of the above influences on the circulation dynamics have to be tested, by considering the individual and combined effects of the assimilation of different types of data.

To begin with the present study, we only considered elementary water properties observations that relatively easily could be obtained from available platforms. The present OSSE only attempts to initiate a first and essential step in the much needed extended studies of advanced modelling as well as data assimilation. We also note that there are not many near-real-time observations available in the Marmara Sea at present; building the necessary infrastructure to incorporate various other types of observations still needs further serious efforts.

The proposed initial observing system should result in better forecasts in terms of water properties of the upper layer, which promises to improve forecasts in the Aegean Sea. To further improve the forecasts, the assimilation of data on water properties, sea level and currents measured by floats and fixed stations (e.g. ADCPs, tide gauges and/or altimeter measurements), the use of these measurements to better estimate volume fluxes through the Bosphorus and Dardanelles Straits would be in order. Appropriate use of such extended measurements tested by continued OSSE's could also impact better estimates of the lower layer circulation driven by the density gradients.

3. *There are many technical details about specific solutions implemented in the data assimilation scheme, but impacts those solutions are not tested in the study.*

While implementing our solutions in this study, we benefited from the experience provided by similar studies in the literature, although we carefully adopted a version needed in our application. We agree with the reviewer that the impact of the present choices and other possible solutions should be tested, possibly by us and/or others in continuing studies in the region.

4. *I think that the style of writing should be improved. It is very difficult to read and interpret many sentences providing important information.*

We will do our best to improve the text in the revised version. The differences will be demonstrated by comparing the discussion paper and revised version.

**Minor comments**

1. *Page 2, line 2: The "high resolution" of what?*

The original phrase is "Until recently, the need for high resolution in the straits made it infeasible to model the complete TSS due to the computational cost."

We clarify the phrase as "Until recently, building a model solving for the hydrodynamics of the complete TSS was not considered a feasible undertaking, implying high computational costs of the required horizontal and vertical resolution enabling to represent the sharp stratification and the extremely complex topography of the straits and shelf regions, largely differing from those of the larger neighboring basins."

2. *Page 2, line 3: What are "integral models"?*

".. integral models of the system ..."

We rephrase the phrase above as "...models of the whole system..."

3. *Page 2, line 12: The OSSE abbreviation is introduced, but it is not explained.*

We change it as "We follow the Observing System Simulation Experiments (OSSE) ..."

4. *Page 3, line 14: same as comment 2.*

"The high complexity of the system requires integral modeling approaches to represent the links between its different compartments."

We replace the above sentence with "The high complexity of the system requires models that simultaneously solve for the whole system in its smallest resolved details and optimally representing multiple scales of interest."

5. *Page 3, lines 19-20: This sentence is not related to the scope of the study.*

We omit the sentence "It was developed by the Alfred Wegener Institute as the first global ocean model using an unstructured mesh."

6. *Page 4, line 11: What is "covariance information" in this context?*

We replace the term "information" with "distribution" to be more precise. The discussion addresses the issues related the prior distribution after resampling the ensemble to compute the covariance. The EAKF preserves the prior covariance distribution during sampling. The ensemble covariance is used to quantify the relation between pairs of state variables or an observation and a state variable in the linear Gaussian context of the ensemble Kalman filter.

7. *Page 4, line 12: What is "prior information" in this context?*

The same response as to comment #6. We replace the term "information" with "distribution"

8. *Page 4, line 13: Which covariances are updated?*

We replace the term with "analysis covariances".

9. *Page 4, lines 20-24: Is vertical diffusion the most important for correctly simulating the depth of the interface? Vertical diffusion should be governed by slowly varying large-scale fields and perturbing it at the high frequency may add processes that may be unphysical.*

The depth of the interface in the TSS is mainly determined by the volume fluxes through the straits at seasonal time scales. On daily time scales, however, the wind forcing may alter the interface depth significantly, especially when there is a severe storm passage over the system (Book et al., 2014). Moreover, enhanced vertical mixing around the interface may contribute to the water exchange between strongly stratified upper and lower layers by entrainment processes (Özsoy et al. 2001).

We agree with the reasoning of the reviewer on vertical diffusion. However, what we perturb is the background vertical diffusivity $K_{v0}$ which is at least two orders of magnitudes smaller than the spatially varying vertical diffusivity $K_v$. We are aware that more care is needed for longer experiments. However, within the period of the experiments, we haven't identified any instability developing due to the growth of the perturbation.

10. *Page 4, line 35: Localization may introduce strong dynamical imbalances. This contrasts the sentence on lines 11-13.*

This is correct. However, since we can only afford 30 ensemble members, localization is required for a system that produces forecasts with small RMSE. The localization half-width radius has been chosen

empirically to minimize RMSE while maintaining a prior RMSE to spread ratio that is approximately unity. A larger ensemble size would allow us to use less stringent localization, further reducing RMSE while also reducing dynamical imbalances. The fact that the assimilation cycle is stable over a number of days suggests that the dynamical imbalance is not large enough to dominate the balanced model dynamics.

11. *Page 6, lines 29-30: I do not understand this sentence.*

"This approach was chosen because FESOM in the Marmara Sea is sensitive to equally plausible salinity boundary conditions."

We prefer to add an appendix to document the properties of the nature run and its difference from the forward model as requested by Reviewer#2. Very briefly, the nature run is supposed to be the best realization of the system whereas forward model is chosen as a different model or the same model with different resolution or different physics. In our study, different surface salinity boundary conditions have been implemented in the two configurations of the same model constituting the forward model and the nature runs. Both solutions appear realistic and equally plausible but different from each other, displaying sensitivity to the applied boundary conditions.

12. *Page 9, line 3: Argo floats and gliders are not instruments.*

Rephrased the following part "it is not easy to deploy instruments such as argo or glider since there is heavy ship traffic." to read as follows: "it is not easy to deploy, for instance, argo floats or gliders close to the surface since …."

13. *Page 9, lines 18-20: How is the sampling rate of 1 minute obtained output frequency of 1 hour.*

In the manuscript, it is explained as "… from hourly NR outputs at varying spatial location along the track of the ferries." That means the synthetic observations falling into the same one hour interval are sampled from the same output but on different locations mimicking the motion of the ferries in time. We modify the above phrase as "…from hourly NR outputs at varying spatial locations so as to remain within one minute time intervals along the track of the ferries."

14. *Page 9, line 24: What is the meaning of stochastic in this context?*

In stochastic EnKF, the observations are perturbed for each ensemble member before assimilation. In this work, the EAKF scheme used in the present work is deterministic since observations are identical for each ensemble member.

15. *Page 9, line 25: Do you want to say that errors of observations are uncorrelated?*

Obviously, repeated observations from the same instrument are expected to have some correlated error component. Here, we neglect the correlated error, as is done in many geophysical assimilation problems, because of the expense and difficulty of dealing with it explicitly. Employing an accurate estimate of the correlated error, or explicitly modeling the correlated error is difficult but would certainly lead to some improvement in our forecast fits to observations. Note that temperature and salinity errors are from unique instruments and might be expected to have less correlated error. Here, temperature and salinity errors are assumed to be uncorrelated and they are set different following an existing observing system presented by Grayek et al. (2011).

16. *Page 10, line 16: Updates are smaller than what? It looks like temperature and salinity are compared by magnitude, but they are two different physical parameters.*

The salinity updates at 20 m. are smaller than the salinity updates in the upper layers, although we refrain from presenting an extra figure in order to reduce clutter. Since the corrections around 20 m. are generally smaller than those at 5 m. for both temperature and salinity and we only choose to show the updates for temperature at 20 m.

17. *Page 11: The bottom paragraph should be reformulated by using the correct terminology.*

We rewrite the paragraph is as:

"The DART offers tools to control the data used without any difficulty. Each type of observation can either be assimilated or withheld to evaluate the resulting analyses. DART also provides a rudimentary quality control capability that can reject observations that are too different from the ensemble mean prior estimate. Synthetic observations are not used in the assimilation (rejected) if $x_b - y > TE(x_b - y)$. Here, $y$ is the observation and $x_b$ is the corresponding prior mean of the ensemble of forward operators. The expected value, E, is computed as:

$$E(x_b - y) = \sigma^2_{xb} + \sigma^2_y \quad (1)$$

where $\sigma$ stands for the standard deviation. T is chosen as 3 for both temperature and salinity in these experiments."

18. *Page 12, lines 1-4: I do not understand why there are outliers in an OSSE experiment? Are observations wrong, or some assumptions are not valid?*

The outliers are completely an outcome of our choice considering the realistic errors in the Marmara Sea. In Aydoğdu et al. (2018), validation of the long-term simulations has been performed and presented. Comparison with the CTD observations shows that the errors may increase up to 3 psu or 3 ℃, for salinity and temperature respectively, in the area of interest but not more. Therefore, we set a limit for the synthetic observations and do not assimilate them if they exceed these error ranges with respect to the prior ensemble mean. The difference between the nature run and ensemble mean can be high depending on the perturbations applied, therefore, there are some observations which we mark as outliers. The approach is chosen to stick with the realistic applications of the data assimilation where a quality check is generally required.

19. *Page 15, line 4: Innovations are better than what?*

We compute the innovation regardless if the observations are assimilated or not. Here, we compare the FB001 and FB002 cases, and conclude that the difference between the synthetic observations and the corresponding ensemble prior mean is smaller in FB002, implying improved forecast as a result of the assimilation.

20. *Fig. 11: Salinity differences have very large gradients. I suspect that they form strong density gradients impacting currents. Currents should be included in the evaluation of the assimilation.*

We added a new figure (Fig. 13) to compare the velocity fields in day 7. We couldn't identify any strong impact of the assimilation on the currents as discussed in our response to major comment #1.

**Authors' response to RC#2**

**Major comments:**

A. *Unless this has been published elsewhere, it would be useful to have a brief analysis of the physical situation in the NR at the time of the assimilation experiments, if possible*

Hopefully, these details will be published in another manuscript that is presently submitted to Ocean Science (Aydoğdu et al. 2018 in the references). Yet, to enable a review of the earlier results, we prefer to provide an appendix using the following figures along with further technical details of the nature run.

[Figure]

**Fig. A1** Daily mean of sea surface temperature (SST) and salinity (SSS) and volume temperature (VT) and salinity (VS) in the Marmara Sea.

[Figure]

**Fig. A2** 5 m. salinity fields in 7 January 2009 at 00:00 simulated in the nature run. The corresponding circulation is overlaid by arrows. Corresponding fields for the FB001 and FB002 experiments are shown in Fig. 13 for comparison.

B. *I had an overarching question in my mind throughout my reading of the manuscript: is the combined effect of i.c. perturbations (a mix of short-term time lag and interannual variability) and diffusivity perturbations able to explain at least part of the model-data differences, within observational error? This question can be posed for*
*(B1) simulated data*

*The B1 question is a question of consistency of the innovations with Ensemble spread + obs error: it is partially covered in the ms. (e.g. Fig.8), but not exploited.*

*Coming back to question B1: I have been frustrated that Fig.8 shows RMSE (of the dimension of the variable) and spread (of the dimension of the variable, \*squared\*). It would have been better to show \*MSE\* (not RMSE) and spread (assuming that this is \*prior\* Ensemble spread): then you could have*

*tested whether the (prior) innovation variance was more or less of the same order as the (prior) "Total spread" (= your estimate of prior error + your estimate of obs error). I did the squaring visually, and the orders of both quantities do not match each other, especially for salinity. I believe that this should even briefly be discussed.*

[Figure]

**Fig. 8** Time series of mean squared innovations, spread and total spread of salinity (top) and temperature (bottom) for FB001 (left) and FB002 (right). The statistics are computed in the location of the observations used in the corresponding assimilation cycle. The spread is the square root of the variance. Total spread is the observational error added to the spread. Y-axis shows the range of each statistics indicated in the legend. Bottom panel of each figure shows the number of available observations (Nposs) in each assimilation cycle. For the assimilation experiment, FB002, Nused and Nout show the number of assimilated observations and outliers, respectively. For the experiment without assimilation, FB001, they are the number of observations which would be assimilated or rejected, respectively, in that specific assimilation cycle if assimilation was performed.

We agree and thank to the reviewer. We checked the calculation of the spread and found out that it is the standard deviation, not the variance. The statistics are accurate but units should be in psu and ºC, for the salinity and temperature respectively. Therefore, they are comparable with the corresponding RMSE. We corrected the mistake on the legend as well as in the caption and provide the revised version of the Fig. 8. The resulting total spread is therefore, the square root of the sum of the variance and observation error. We will revise the manuscript accordingly.

The ensemble spread is initially provided by the perturbation of the initial temperature and salinity fields and maintained by the background vertical diffusivity perturbation. However, the growing RMSE points out different salinity boundary conditions chosen for the forward model and the nature run. In another word, the perturbations maintaining the spread are capable of sustaining it even the observations are assimilated. On the other hand, assimilation will significantly reduce the errors due to the physical processes simulated by chosen model schemes. We will revise the manuscript accordingly.

*(B2) any existing real observations (e.g., SST).*
*The B2 question is about the realism of errors: it was not covered in the ms (only the conclusion mentions "lack of data").*

The error assessment of the model using CTD observations is performed in the Marmara Sea and presented in another study (Aydoğdu et al 2018). We think the errors in the present work compare well with those and are therefore expected to be realistic. Lack of data is mentioned only for data assimilation purposes. We will summarize this aspect in the appendix that we shall provide according to what has been implied in major comment #A.

*C. Why did you limit yourself to 7 days? Some of the error processes, especially those associated with mixing and stratification, could act on longer time scales.*

The main challenge to perform longer experiments is commensurate with the computational cost of running ensembles using such a high-resolution model. However, given the present results, we are motivated to perform longer experiments which take into account also the suggestions addressed by the reviewers.

*D. The localization cut-off scales are very short. Can't this trigger fast unphysical responses, for instance via temperature-sea level covariances?*

Agree, that is possible. However, the cut-off radius has been chosen to minimize RMSE while maintaining a prior RMSE to spread ratio that is approximately unity. On the other hand, the spurious correlations related to larger cut-off radius may also trigger unphysical behaviors in energetic basin such as the Marmara Sea. But we haven't identified any unphysical behavior within the period of the experiments related to dynamical imbalances.

**Minor comments:**

*1. Page 4, lines 11-12: How is "prior covariance information" related to "dynamical balances?*

We replace the term "information" with "distribution" to be more precise. The ensemble covariance is the statistical representation of the prior constraints ('dynamical balances') generated by the model. For a linear model, a sufficiently large ensemble would be able to represent all the balances. However, our model is nonlinear and cost forces us to use a small ensemble. This means that the ensemble sample covariances do not exactly represent the dynamical balances. The ensemble filter algorithm and inflation we use are designed to maintain the prior ensemble covariance as much as possible. The benefits of preserving the prior distribution are extensively discussed in Anderson (2001) compared to ensemble filters such as kernel or Gaussian resampling filters. However, given the errors in the sample covariances, localization is effective in reducing forecast RMSE compared to observations. Localization also disrupts the prior sample covariance, so can result in reduced dynamical balance.

*2. Page 4, lines 11 and 13: "the covariances are updated in every assimilation cycle": Isn't this contradictory with "it preserves the prior covariance information"?*

We replace the term information with distribution. Analysis covariances are updated in every assimilation cycle.

*3. Page 4, line 25: The generally adopted procedure to perturb diffusivity parameters does not use a centered Gaussian pdf*

We were not aware of it. We thank to the reviewer for the information and will further dig in the literature if we use the same perturbation methodology.

*4. Page 14, line 5: "is similar" -> "behaves similarly"*

Agreed.

*5. Page 15, lines 8-9: "...correct the subsurface fields": Fig. 9 is in data space (surface), so one cannot see a subsurface effect from that figure.*

We replace the term 'subsurface' with 'water column above the pycnocline, which is actually meant there.

**Authors' response to editor' comments**

*1. P. 2, ll. 26-28. The depth of the transition between the two layers is not mentioned (see also Fig. 4 and associated comments).*

We add the depth of transition to the end of the sentence "The exchange of contrasting water masses forms a highly stratified water column structure throughout the system ..." as "with a pycnocline around 20 m. depth."

*2. Fig. 7. What are exactly the increments shown there (I understand there is one set of increments for each element of the assimilation ensemble?*

The ensemble mean increment and innovation of the ensemble are used throughout this study.

*3. Eq. (1). I understand sigma2y is the assumed variance of the observational error. But what exactly is sigma2xb ? (I presume it is defined from the prior ensemble. But how exactly?). Anyway, you must also respond to minor comment 18 of referee 1.*

$\sigma^2_{xb}$ is the variance of the prior ensemble estimate to the observation y (the forward operator ensemble). We reformulated the paragraph as requested by reviewer #1 in minor comment 17.

*4. The spatial area over which the diagnostics shown in Fig. 8 have been computed does not seem to be mentioned.*

We add to the caption the sentence "The statistics are computed in the location of the observations used in the corresponding assimilation cycle."

*5. I presume most of the readers (but maybe not all ...) will easily locate the Bosphorus and the Dardanelles on the various maps. But it may not be the same as concerns the Gulf of Izmit (p. 16, l. 11).*

Agreed. We can name the region as it as 'the northeastern Marmara Sea' and will mark locations on one of the Figures.

**References**

Anderson, J. L. (2001). An ensemble adjustment Kalman filter for data assimilation. Monthly weather review, 129(12):2884–2903.

Aydoğdu, A., Pinardi, N., Özsoy, E., Danabasoglu, G., Gürses, Ö., and Karspeck, A. (2018) Circulation of the Turkish Straits System between 2008–2013 under complete atmospheric forcings, Ocean Sci. Discuss., https://doi.org/10.5194/os-2018-7, (submitted, in review).

Book, J. W., Jarosz, E., Chiggiato, J., and Beşiktepe, Ş. (2014). The oceanic response of the Turkish Straits System to an extreme drop in atmospheric pressure. Journal of Geophysical Research: Oceans, 119(6):3629–3644.
Grayek, S., Staneva, J., Schulz-Stellenfleth, J., Petersen, W., and Stanev, E. V. (2011). Use of FerryBox surface temperature and salinity measurements to improve model based state estimates for the German Bight. Journal of Marine Systems, 88(1):45–59.

Özsoy, E., Di Iorio, D., Gregg, M. C., and Backhaus, J. O. (2001). Mixing in the Bosphorus Strait and the Black Sea continental shelf: observations and a model of the dense water outflow. Journal of Marine Systems, 31(1):99–135.

---

## Editor Decision (ED1)

Dear Ali,

I have now received comments, from two referees, on the latest version of your paper. The referees are the same as those of the previous version. Referee 1 had asked for a major revision and a further review. Referee 2 had only requested minor revisions, without further review. I have nevertheless sent the new version to him/her, asking him/her to check if the revisions he/she has asked for had been properly made.

Both referees consider the paper can be published as it is. I therefore accept it, but nevertheless ask you to make a few final editing corrections.

1. Please define HCMR (p. 9, l. 5 ) and OSE (p. 18, l. 13).
2. Caption of Fig. 8, l. 3. *Y-axis* → *vertical axis*.
3. Figure 8 and associated text. You mention *Nout*, while what is written on the figure itself seems to be *Nbdd*. Please correct.
4. P. 15, l. 12, *… for the exemplary case on …* → *… on the example of …* (that is what I understand ; the meaning is different)
5. Caption of Fig. A1. Define more precisely what *volume temperature* and *volume salinity* are.

I thank you again for having submitted your paper to the NPG Special Issue in tribute to Anna Trevisan,

Olivier Talagrand
Editor, NPG